# COVID-19 Vaccines: Where Did We Stand at the End of 2023?

**DOI:** 10.3390/v16020203

**Published:** 2024-01-29

**Authors:** Kenneth Lundstrom

**Affiliations:** Pan Therapeutics, 1095 Lutry, Switzerland; lundstromkenneth@gmail.com

**Keywords:** inactivated and live whole-virus vaccines, protein and peptide vaccines, viral vector vaccines, DNA vaccines, RNA vaccines

## Abstract

Vaccine development against SARS-CoV-2 has been highly successful in slowing down the COVID-19 pandemic. A wide spectrum of approaches including vaccines based on whole viruses, protein subunits and peptides, viral vectors, and nucleic acids has been developed in parallel. For all types of COVID-19 vaccines, good safety and efficacy have been obtained in both preclinical animal studies and in clinical trials in humans. Moreover, emergency use authorization has been granted for the major types of COVID-19 vaccines. Although high safety has been demonstrated, rare cases of severe adverse events have been detected after global mass vaccinations. Emerging SARS-CoV-2 variants possessing enhanced infectivity have affected vaccine protection efficacy requiring re-design and re-engineering of novel COVID-19 vaccine candidates. Furthermore, insight is given into preparedness against emerging SARS-CoV-2 variants.

## 1. Introduction

During the last 3 years, the global population has been exposed to the serious COVID-19 pandemic followed by tragic loss of lives, lockdowns, and social and economic consequences [1]. The rapid development of and mass vaccinations with efficient COVID-19 vaccines [2] contributed to the downgrading of the pandemic to an endemic status in the spring of 2023 [3]. However, the emergence of more infectious and potentially pathogenic SARS-CoV-2 variants [4] has further presented new challenges to achieving a complete eradication of the disease.

The extensive and rapid spread of SARS-CoV-2 triggered the development of COVID-19 vaccines on a broad front of vaccines based on inactivated and attenuated whole viruses, protein subunits and peptides, viral vectors, DNA, and RNA [2]. The unprecedented collaboration of academic institutions, pharmaceutical and biotechnology companies, and governmental authorities provided the safety and efficacy of different types of COVID-19 vaccines in animal models and clinical trials. It led to the granting of emergency use authorization (EUA) in a number of countries around the world. In this review, the different types of COVID-19 vaccines are described based on their composition, efficacy, and potential relationship to post-vaccination adverse events. Moreover, the effect of SARS-CoV-2 variants of concern (VoC) related to vaccine efficacy and the re-engineering to address immune escape by VoC are presented. Finally, strategies for future approaches to address emerging variants and other future viral infections are discussed. 

## 2. COVID-19 Vaccines

The urgent need for efficient vaccines against SARS-CoV-2 has resulted in parallel development including classic whole-virus vaccines and more novel approaches based on nucleic acids. As these approaches are significantly different, it is appropriate to describe them separately below. The COVID-19 vaccine development is further summarized in Table 1. 

### 2.1. Inactivated and Attenuated Whole-Virus Vaccines

The classic whole-virus vaccine development comprises the application of inactivated or live attenuated whole virus [5]. One aspect of applying whole viruses for vaccine development is the possibility of inducing pan-immune responses, which are not restricted to targeting only the commonly used spike (S) protein but also the matrix, envelope, and nucleoproteins of SARS-CoV-2. Another essential feature is that whole-virus vaccines can be stored at 2–8 °C. This facilitates the distribution and administration in developing countries [6]. In the case of inactivated whole-virus vaccines, the SARS-CoV-2 cultured in cells can be inactivated by formaldehyde or glutaraldehyde treatment or UV or gamma radiation [7]. The downside of using whole viruses relates to the production of contagious virus particles and the potential degradation of viral antigens and epitopes caused by the inactivation process [7,8]. Several inactivated whole-virus COVID-19 vaccines have been developed [7] (Table 1).

The COVI-VAC vaccine is based on a single-dose intranasal administration of a live attenuated SARS-CoV-2 carrying the S protein with 283 silent mutations [9]. COVI-VAC has shown strong induction of immune responses and long-lasting cellular immunity [9]. Moreover, good tolerability and safety and a seroconversion rate of 83% were achieved in a phase I/II trial [10]. The two-dose CoronaVac (Sinovac) aluminum (Alum) hydroxide-adjuvanted whole-virus vaccine is inactivated by β-propiolactone (BPL) treatment [11]. It showed good seroconversion and immune memory, causing only minor or moderate adverse events in phase I and II trials [11,12]. In phase III, CoronaVac provided 50.7% and 100% efficacy against symptomatic COVID-19 and hospitalization, respectively [13]. In June 2021, CoronaVac was granted EUA in 54 countries [14]. Another BPL-inactivated whole-virus vaccine, VLA2001, which is adjuvanted with Alum and CpG 1018.31 adjuvants, elicited robust antibody responses in preclinical studies [15]. It showed good safety and dose-dependent humoral and cellular responses in a phase I/II study [16]. The VLA2001 was superior to the adenovirus-based ChAdOx1-S vaccine in phase III [17]. EUA was granted for VLA2001 in Bahrain in March 2022 [18].

The Vero cell-based ERUCOV-VAC vaccine provided 100% protection in K18-hACE2 mice challenged with SARS-CoV-2 [19]. Moreover, acceptable immunogenicity and safety were established in phase I/II in healthy volunteers vaccinated with ERUCOV-VAC [20]. In phase III, interim results demonstrated a 41.03% reduced risk of COVID-19 compared to the CoronaVac vaccine [21]. The Alum-adjuvanted COVIran Barekat vaccine showed efficient protection against SARS-CoV-2 in rhesus macaques [22]. In a cohort study, the COVIran Barekat vaccine demonstrated an 85% reduction in mortality and a significant reduction in hospital admissions [23]. The Indian Covaxin (BBV152) inactivated whole-virus vaccine elicited strong immunogenicity and protected rhesus macaques against challenges from SARS-CoV-2 [24]. Moreover, a 93.4% vaccine efficacy was seen in a phase II trial [25]. Good safety and efficacy against symptomatic COVID-19 were reported in phase III [26] for BBV152. Another whole-virus vaccine, the Alum hydroxide-adjuvanted and formalin-inactivated QazCovid-in, protected hamsters against SARS-CoV-2 challenges [27]. It showed 100% and 92–94% seroconversion in phase I and phase II studies, respectively [28]. In phase III, an 82% vaccine efficacy was achieved after two doses of QazCovid-in [29].

The inactivated BBIBP-CorV (Sinopharm) whole-virus vaccine, based on a SARS-CoV-2 isolate from a COVID-19 patient in China [30], induced strong neutralizing antibody responses in several animal models and protected rhesus macaques from SARS-CoV-2 challenges [30]. In phase I, BBIBP-CorV showed good safety and tolerability and induced robust humoral responses and high titers of neutralizing antibodies [31]. A vaccine efficacy of 78.1% was reported in phase III [32]. The BBIBP-CorV vaccine was granted EUA in China in July 2020 [33] and by the WHO in May 2021 [34].viruses-16-00203-t001_Table 1Table 1Examples of different COVID-19 vaccines in clinical trials.Type of VaccineVaccine DesignStageObservations**Whole-virus**


COVI-VAC(Codagenix)Intranasal delivery of mutated attenuated whole virusI/IILong-lasting immunity [9], 83% seroconversion [10]CoronaVac(Sinovac)Two-dose BPL-inactivated Alum hy-droxide whole virusI/IIIIIImmune memory, seroconversion [11,12]Good efficacy against COVID-19 [13]VLA2001(Valneva)Alum- and CpG 1018.31-adjuvanted, BPL-inactivated whole virusI/IIIIIEUAHumoral and cellular immunity [16]Superior Ad vector-based vaccine [17]Approval in Bahrain in March 2022 [18]ERUCOV-VACVero cell-based inactivated whole virusI/IIIIISafety and immunogenicity [20]Superior to CoronaVac [21]COVIran Barekat(BIV1 CovIran)BPL-inactivated, Alum-adjuvanted whole virusCS85% reduction in mortality [23]Covaxin(BBV152)Inactivated whole-virus vaccine based on the SARS-CoV-2 NIV-2020-770 strain I/IIIII93.4% vaccine efficacy [25]Efficacy against symptomatic COVID [26]QazCovid-inFormalin-inactivated, Alum hydroxide-adjuvanted whole virusIIIIII100% seroconversion [28]92–94% seroconversion [28]82% vaccine efficacy [29]BBIBP-CorV(Sinopharm)Inactivated whole-virus vaccine based on SARS-CoV-2 HB02 patient isolateIIIIEUAHumoral and nAb responses [31]78.1% vaccine efficacy [32]Approval in China [33], by the WHO [34]**Protein and peptide**


preS dTMBaculovirus-expressed purified prefusion-stabilized S protein + AS03 adjuvantIILower-than-expected immunogenicity [35]Good safety, robust immunogenicity [36]SCB-2019AS03- or CpG/Alum-adjuvanted S TrimerIII/IIIHumoral and cellular responses, nAbs [37]Immunogenicity, cross-reactivity [38]COVAX-19ECD of SARS-CoV-2 adjuvanted with Alum-CpG55.2 or Advax-CpG55.2IIIIISafety, humoral and cellular responses [39]Reduction in COVID-19 and its severity [40]NanocovaxAlum hydroxide-adjuvant SARS-CoV-2 S I/IIIIISafety, tolerability, and immunogenicity [41] 90% vaccine efficacy [42]Razi Cov ParsIntranasal RAS-01 adjuvanted S TrimerIIHumoral and cellular immunogenicity [43]MVC-COV1901CpG 1018 and Alum hydroxide adjuvanted prefusion stabilized S TrimerIIIIIInAbs, enhanced immunogenicity [44]Safety, promising immune responses [45]Safety, robust immunogenicity [46]EpiVarCorona(EVCV)Chemically synthesized peptide antigens of SARS-CoV-2 SI/IIIIISafety, prevention of COVID-19 [47]82.5% vaccine efficacy [48]SCTV01CTrimeric extracellular SARS-CoV-2 S domain adjuvanted with squaleneIIIIEUARobust immune responses [49]Confirmed strong immunogenicity [50]Approved in China [51]ZF2001Tandem RBD SARS-CoV-2 S repeat with alum hydroxide adjuvantI/IIIIIEUAnAb in 83–97% of vaccinees [52]Protection against COVID-19 [53]Approval in China [54] and Uzbekistan [55]CIGB-66(Abdala)RBD SARS-CoV-2 expressed in yeast adjuvanted with alum hydroxideIIIIEUASafety, robust immune response [56]92.3% vaccine efficiency [57]Approval in Cuba [58]BECOV2(Corbevax)RBD SARS-CoV-2 expressed in yeast adjuvanted with alum and CpGI/IIIIIEUAHigh nAb titers [59]High nAB titers, low adverse events [60]Approval in India [61] and Botswana [62]FINLAY-FR-2RBD SARS-CoV-2 coupled to tetanus toxoid, adjuvanted to alum hydroxideI/IIIIIElevated immunogenicity [63]Good protection against COVID-19 [64]UB-612RBD fused to sc IgG1 Fc, 5 Th/CTL epitopes + alum phosphate adjuvantI/IILong-lasting nAb responses, broad T-cell immunity [65].ReCoVTwo-component NTD-RBD subunit vaccine + BFA03 adjuvantIIEUASuperior to mRNA vaccines [66]Approval in Mongolia in March 2023 [67]NVX-CoV2373NP-encapsulated full-length SARS-CoV-2 S + Matrix-M1 adjuvantI/IIIIICMARobust immunogenicity, Th1 biased [68]100% protection against severe disease [69]Approval in the EU and GB [70]GBP510Self-assembled NP vaccine adjuvanted with AS03I/IIIIITolerability, immunogenicity [71]Superior to ChAdOx1-S [72]**Viral vectors**


ChAdOx1 nCoV-19Ad-based full-length SARS-CoV-2 SI/IIIIIEUARobust humoral and cellular responses [73]62–90% vaccine efficacy [74]Approval in the UK in December 2020 [75]Ad5-nCoVAd5-based full-length SARS-CoV-2 SIIIIIIEUABinding and nAb responses [76]Dose- and age-dependent [77]Good safety and efficacy [78]Approval in China in February 2021 [79]Ad26.COV2.SAd26-based prefusion-stabilized SARS-CoV-2 S using a single immunizationI/IIIIIEUARobust immune responses [80]52.9% vaccine efficacy [81]Approval in the US in February 2021 [82]rAd26S/rAd5-S(Sputnik V)Prime vaccination with Ad26 SARS-CoV-2 S followed by Ad26 SARS.CoV-2 S boosterI/IIIIIEUAHumoral and cellular responses [83,84]91.6% vaccine efficacy [85]Approval in Russia in July 2020 [86]GRAd-COV2GRAd-based prefusion-stabilized SARS-CoV-2 SI97.7–100% seroconversion rates [87]MVA-COV2-SVV MVA strain expressing SARS-CoV-2 SIITrial in progress [88]NDV-HXP-SNDV expressing SARS-CoV-2 SISafe, potent immunogenicity [89]LV-SMENP-DCTransduced LV expressing SARS-CoV-2 structural proteins and proteaseI/IITrial in progress [90]IFV dNS1-RBDCold-attenuated influenza virus strain with NS1 deletion and RBD insertionI/IIWeak T-cell responses, modest humoral and mucosal immune responses [91]MV-COV-2 S(TMV083)MV-based expression of full-length SARS-CoV-2 SITolerability, inferior immune response compared to COVID-19 patients [92]VSV-SARS-CoV-2 S(V590)VSV-based expression of SARS-CoV-2 SIGood safety and tolerability but weak immunogenicity, termination of trial [93] VSV-ΔG(IIBR-100)Chimeric VSV where VSV-G replaced by SARS-CoV-2 SIIRobust nAb responses [94]**Nucleic acid—DNA**


INO-4800DNA-based full-length SARS-CoV-2 SIIICellular, humoral immune responses [95]Good safety and tolerability [96]ZyCoV-DDNA-based SARS-CoV-2 S RBDI/IIIIIEUARobust immune responses [97]66.6% vaccine efficacy [98]Approval in India in 2021 [99]GX-19Synthetic SARS-CoV-2 S RBDIGood safety and tolerability, superior immunity of GX-19 compared to GX-19N [100]GX-19NS RBD Foldon, N and S proteins**Nucleic acid—RNA**


BNT162b(Pfizer/BioNTech)Prefusion-stabilized full-length SARS-CoV-2 S RNA in NPsI/IIIIIEUARAGood safety and immunogenicity [101]95% vaccine efficacy [102]Approval in the EU and CH in 2020 [103]Approval by the FDA in August 2021 [104]mRNA-1273(Moderna)Prefusion-stabilized full-length SARS-CoV-2 S RNA in LNPs IIIIEUAS-specific immune responses [105]94.1% vaccine efficacy [106]Approval in the US in December 2020 [107]CVnCOV(Curavec)Full-length SARS-CoV-2 S RNA in LNPsIIIIS-specific immune responses [108]48.1% vaccine efficacy [109]RCoVThermostable LNP-encapsulated SARS-CoV-2 S RBD RNAIHumoral and cellular responses [110]LNP-nCoVsaRNALNP-encapsulated VEE-based replicon expression full-length SARS-CoV-2 SIIISafe, <100% seroconversion [111]Prime-booster: >seroconversion [112]BPL, β-propiolactone; CH, Switzerland; CMA, conditional marketing authorization; CS, cohort study; ECD, extracellular domain; EU, European Union; EUA, emergency use authorization; GB, Great Britain; GRAd, gorilla adenovirus; LNP, lipid nanoparticle; LV, lentivirus; MVA, modified vaccinia Ankara strain; nAbs, neutralizing antibodies; NDV, Newcastle disease virus; NP, nanoparticle; NTD, N-terminal domain; RA, regulatory approval; RBD, receptor binding domain; S, spike protein; UK, United Kingdom; US, United States; VEE, Venezuelan equine encephalitis virus; VSV, vesicular stomatitis virus; VV, vaccinia virus; WHO, World Health Organization.

### 2.2. Protein Subunit and Peptide Vaccines

In the case of protein subunit and peptide vaccines, three main approaches have been explored, comprising the recombinant SARS-CoV-2 S protein, its receptor binding domain (RBD), and nanoparticle formulations for vaccine development [112], as described below and summarized in Table 1.

Several vaccine development approaches have involved purified recombinant prefusion-stabilized and transmembrane-deleted SARS-CoV-2 S expressed in insect cells from baculovirus vectors [113]. For example, protection against SARS-CoV-2 was demonstrated in non-human primates after immunization with the AS03-adjuvanted preS dTM vaccine [113]. In phase I, although good safety was obtained, lower-than-expected immunogenicity suggested that additional antigen formulation and dose optimization was required [35]. Two doses of the preS dTM-AS03 vaccine showed acceptable safety and elicited strong immune responses in phase II [36]. The AS03- or CpG/alum-adjuvanted SCB-2019 vaccine comprising the trimeric S protein (STrimer) provided protection against SARS-CoV-2 in immunized non-human primates [114]. Strong humoral and cellular responses and high titers of neutralizing antibodies were obtained in a phase I study [37]. In phase II/III, SCB-2019 elicited immunogenicity and cross-reactivity against SARS-CoV-2 [38].

In the COVAX-19 vaccine, the extracellular domain (ECD) of SARS-CoV-2 S was adjuvanted with Alum-CpG55.2 or Advax-CpG55.2, which induced robust humoral and cellular immune responses in mice and demonstrated protection against SARS-CoV-2 challenges in hamsters [115]. In phase II, COVAX-19 showed good safety and strong humoral and cellular immunogenicity [39]. Additionally, significantly reduced COVID-19 rates and severity of disease were reported for COVAX-19 in phase III [40]. The Alum hydroxide-adjuvanted Nanocovax vaccine expressing SARS-CoV-2 S generated neutralizing antibodies in mice, hamsters, and macaques [116]. Good safety, tolerability, and robust immune responses were demonstrated in phase I/II [41]. Moreover, a 90% vaccine efficacy against the SARS-CoV-2 Wuhan strain was established in phase III [42].

In another approach, intranasal administration of two doses of the Razi Cov Pars SARS-CoV-2 S protein subunit vaccine adjuvanted with RAS-01 elicited robust humoral and cellular responses and protected hamsters from SARS-CoV-2 challenges [117]. Moreover, durable humoral and cellular immunogenicity was obtained in phase II [43]. The CpG 1018 and Alum hydroxide-adjuvanted prefusion-stabilized S Trimer MVC-COV1901 vaccine reduced lung pathology and protected hamsters against SARS-CoV-2 challenges [118]. Strong neutralizing antibody responses were obtained in phase I [44]. A good safety profile and promising immunogenicity were seen in phase II [45]. These findings were confirmed in a phase III trial in Paraguay [46]. The EpiVarCorona (EVCV) vaccine is based on synthesized peptide antigens and immunogens from the SARS-CoV-2 S protein, which elicited immune responses in various animal models and prevented pneumonia in hamsters and non-human primates [119]. Moreover, good safety and prevention of COVID-19 was observed in phase I/II [47]. An 82.5% vaccine efficacy was reported in phase III [48]. In the case of the bivalent SCTV01C vaccine, the trimeric extracellular domain of SARS-CoV-2 S adjuvanted with squalene-based oil-in-water immunization of mice provided protection against SARS-CoV-2 challenges [120]. The SCTV01C vaccine showed promising immune responses in phase I [49], which was confirmed in phase III [50]. EUA was granted for SCTV01C in China [51].

Among recombinant RBD vaccines, the ZF2001, which is based on tandem RBD repeats of Alum hydroxide-adjuvanted SARS-CoV-2 S, protected mice and non-human primates from SARS-CoV-2 challenges [121]. In phase I/II, neutralizing antibodies were detected in 83% of vaccinees after two doses and in 97% after three injections [52]. In phase III, immunization with ZF2001 provided protection against symptomatic and severe-to-critical COVID-19 [53]. ZF2001 was granted EUA in China [54] and Uzbekistan [55] in March 2021. Another RBD SARS-CoV-2 S-based subunit vaccine, CIGB-66 (Abdala), expressed from *Pichia pastoris* yeast cells, was adjuvanted with Alum hydroxide [122]. CIGB-66 induced robust immune responses in mice and primates [122]. Good safety, tolerability, and robust immune responses were obtained in phase I [56]. A 92.3% vaccine efficacy was demonstrated in phase III [57]. EUA was granted for CIGB-66 in Cuba in July 2021 [58]. In another approach, the BECOV2 vaccine based on the expression of the RBD in yeast cells adjuvanted with Alum and CpG deoxynucleotide elicited high levels of neutralizing antibodies and T-cell immunity in mice [123]. High neutralizing antibody titers were also obtained for BECOV2 (Corbevax) in phase I/II [59]. Moreover, in comparison to the Covishield vaccine, Corbevax showed superior immunogenicity and 50% fewer adverse events in phase III [60]. Corbevax has received EUA in India [61] and Botswana [62]. 

In another approach, the SARS-CoV-2 RBD was chemically coupled to the tetanus toxoid and adjuvanted with Alum hydroxide at the Finlay Institute in Cuba as the FINLAY-FR-2 vaccine candidate [124]. Immunization of BALB/c mice induced strong immune responses [124]. In phase I/II, two doses of FINLAY-FR-2 elicited comparable humoral immune responses to natural SARS-CoV-2 infections [63]. Elevated levels of immune responses were obtained in children receiving a booster immunization [63]. In phase III, 76.8% protection against severe COVID-19 and 77.7% against hospitalization was achieved after two doses of FINLAY-FR-2 [64]. Furthermore, a booster vaccination increased the percentage to 96.6 for severe disease and hospitalization [64].

The UB-612 subunit protein peptide vaccine was engineered to contain the RBD fused to a single-chain human IgG1 Fc, five synthetic peptides covering helper and cytotoxic T lymphocyte (Th/CTL epitopes from S2, membrane (M), and nucleocapsid (N) of SARS-CoV-2, and one universal Th peptide plus an Alum phosphate adjuvant) [125]. Robust immune responses were obtained in several rodent species, and protection against SARS-CoV-2 challenges was achieved in mice and non-human primates [125]. UB-612 elicited long-lasting neutralizing antibody responses and broad T-cell immunity in phase I/II [65].

The ReCoV vaccine comprises the trimeric two-component N-terminal domain (NTD) and RBD subunits and a novel adjuvant (BFA03) for binding to the ACE2 receptor. It elicited robust immune responses and protected mice and macaques against SARS-CoV-2 challenges [126]. In phase II in the Philippines, a comparison to mRNA vaccines demonstrated favorable safety and superior immunogenicity of the ReCoV vaccine [66]. Moreover, EUA was granted for ReCoV in Mongolia in March 2023 [67].

Among nanoparticle-based protein subunit vaccines, the NVX-CoV2373 containing the full-length SARS-CoV-2 S and the Matrix-M1 adjuvant showed protection against SARS-CoV-2 in rats and baboons [127]. In comparison to sera from convalescent COVID-19 patients, NVX-CoV2373 induced superior immune responses and CD4^+^ T-cell responses biased towards Th1 in phase I/II [68]. The NVX-CoV2373 demonstrated 100% efficacy against severe COVID-19 and 76.3% efficacy against symptomatic disease in phase III [69]. Conditional marketing authorization was granted in the EU in December 2021 and in Great Britain in February 2022 [70]. 

The GBP510 vaccine is based on self-assembled nanoparticles adjuvanted with AS03 [128]. In phase I/II, GBP510 showed good tolerability and high immunogenicity [71]. In phase III, superior neutralizing antibody titers and seroconversion rates were obtained for GBP510 compared to the adenovirus-based ChAdOx1-S vaccine [72]. 

### 2.3. Viral Vector Vaccines

Much attention has been paid to viral vector-based COVID-19 vaccines where the aim has been to efficiently deliver and express SARS-CoV-2 proteins or their epitopes as antigens [129,130]. The S protein has been the most frequently used target for the expression from a large spectrum of viral vectors such as adenoviruses (Ad), vaccinia viruses (VV), Newcastle disease virus (NDV), measles viruses (MV), rhabdoviruses (RABV), alphaviruses, lentiviruses (LV), and influenza viruses. Various viral vector-based vaccine candidates are described below and summarized in Table 1.

Ad vectors have been most frequently used and have provided the best progress, so far. For example, the ChAdOx1 nCoV-19 vaccine, based on a chimpanzee Ad vector expressing the full-length S protein, has demonstrated robust immune responses in several species and protection against SARS-CoV-2 in rhesus macaques [131]. In phase I/II, humoral and cellular immune responses were obtained [73]. Depending on dosing, a vaccine efficacy of 62–90% was achieved in phase III [74]. The ChAdOx1 nCoV-19 was granted EUA in the UK in December 2020 [75]. The third-generation human Ad5 serotype has been utilized for the expression of the S protein providing protection against SARS-CoV-2 in immunized mice, ferrets, and macaques [132,133]. Both binding and neutralizing antibody responses were detected in volunteers in phase I [134]. However, the immune responses were dependent on pre-existing Ad5 antibodies and the age of the vaccinee, as demonstrated in phase II [76]. Good safety and efficacy were achieved in phase III [77]. EUA was granted to Ad5-nCoV in February 2021 in China [78]. Application of the Ad26 serotype for the expression of the prefusion-stabilized S protein (Ad26.COV2.S) [135] required only a single immunization to elicit robust immune responses and provide protection against SARS-CoV-2 in non-human primates [136]. In phase I/II, Ad26.COV2.S was safe and induced strong immune responses [79]. In phase III, a 52.9% vaccine efficacy was achieved after a single injection [80]. EUA was granted in the US in February 2021 [81]. In another approach with the aim of limiting pre-existing Ad immunity, the SARS-CoV-2 S was expressed from Ad26 and Ad5 vectors. The Sputnik V vaccine strategy comprises prime immunization with rAd26-S followed by booster immunization with Ad5-S [137], which has been indicated to elicit strong immune responses in animal models although no results have been published. Good safety, robust humoral and T-cell-mediated immune responses, and vaccine efficacy were reported in phase I/II [82,83]. Moreover, a 91.6% vaccine efficacy was obtained in phase III [84]. Quite controversially, the Sputnik V vaccine was granted EUA in Russia in July 2020 after only an interim vaccine evaluation was carried out in 76 volunteers and no phase III data were available [85]. Recently, the GRAd gorilla adenovirus has been applied for the expression of the prefusion-stabilized SARS-CoV-2 S protein, which elicited robust immune responses in mice and macaques [138]. In phase I, good safety and 97.7 and 100% seroconversion rates were obtained in young volunteers (18–55 years old) and older adults (65–85 years old), respectively [86]. 

Among vaccinia viruses, poxviruses such as the modified vaccinia Ankara (MVA) strain have been utilized for the expression of the SARS-CoV-2 S and N proteins [139]. Immunization of mice with MVA-COV2-S elicited antigen-specific humoral and cellular immune responses [140]. Additionally, immunization of C57NL/6 mice was well tolerated and induced S-specific humoral and cellular immune responses [141]. In clinical settings, the recruitment for phase I evaluation of MVA-COV2-S has been described [142]. Currently, a phase II trial for MVA-COV2-S (MVA-SARS-2-S) is in progress [87].

The oncolytic Newcastle disease virus (NDV) has also been engineered for the expression of the SARS-CoV-2 S protein [143]. Immunization of mice and hamsters induced strong binding and neutralizing antibodies and protected against SARS-CoV-2 [143]. In phase I, the inactivated SARS-CoV-2 S expressing NDV-HXP-S adjuvanted with CpG 1018 showed acceptable safety and potent immune responses [88].

Lentiviruses (LVs) belonging to the retrovirus family have been utilized for the expression of the full-length SARS-CoV-2 S protein [144]. After systemic administration to ACE2-humanized mice, robust immune responses were obtained. However, only partial protection against challenges with SARS-CoV-2 was achieved [144]. In contrast, intranasal LV-SARS-CoV-2 S administration was superior, providing a significant decrease in viral load in the lung and reduced local inflammation [144]. Lung injury was also prevented in hamsters immunized with LV-SARS-CoV-2 S. In another approach, LV vectors have been engineered for the expression of conserved domains of the SARS-CoV-2 structural proteins and the protease using the SMENP minigenes [145]. Dendritic cells (DCs) transduced with LV-SMENP and subcutaneously administered to antigen-presenting cells (APCs) induced immune responses. A phase I/II trial with LV-SMENP-DC is in progress [89].

Engineering of a chimeric influenza virus vector where the neuraminidase (NA) gene was replaced with a membrane-anchored form of the SARS-CoV-2 S RBD elicited robust neutralizing antibody responses in intranasally immunized mice [146]. In phases I and II, a dNS1-RBD vaccine comprising a cold-adapted influenza virus strain without its non-structural protein 1 (NS1) with an insertion of the SARS-CoV-2 S RBD was engineered [90]. In phases I and II, only weak T-cell responses and modest humoral and mucosal immune responses were induced [90].

Measles viruses (MV) belong to self-amplifying RNA (saRNA) viruses, which have been applied for the expression of the full-length SARS-CoV-2 S protein [147]. Immunization of mice with MV-SARS-CoV-2 S induced Th1-biased antibody and T-cell responses [147]. In phase I, the MV-SARS-CoV-2 S (TMV083) vaccine showed good tolerability [91]. However, the immune responses were inferior to those seen in convalescent COVID-19 patients, which led to premature discontinuation of the study [91]. 

Vesicular stomatitis virus (VSV), a member of the rhabdovirus family, has also been used for SARS-CoV-2 S protein expression [148]. It resulted in the protection of BALB/c mice against SARS-CoV-2 [148]. The VSV-SARS-CoV-2 S (V590) vaccine showed good safety and tolerability in phase I [92]. Disappointingly, the inferior immune responses compared to those detected in convalescent COVID-19 patients resulted in the termination of the trial. In an alternative approach, the VSV G protein was replaced by the SARS-CoV-2 S protein in a chimeric vector (VSV-ΔG) [149]. Administration of VSV-ΔG showed potent neutralizing antibody responses in golden Syrian hamsters [149]. In phase II, robust neutralizing antibody responses against SARS-CoV-2 were obtained after immunization with the VSV-ΔG vaccine [93]. Moreover, a single-dose rhabdovirus vaccine has been engineered, where a chimeric minispike comprising the RBD domain linked to a rabies virus (RABV) transmembrane stem-anchor sequence was engineered [150]. VSV replicons generated cell surface expression of antigen and incorporation into the envelope of secreted non-infectious particles. A single administration of VSV replicons complemented with VSV G (VSV-ΔG-minispike-eGFP) protected transgenic K18-hACE2 mice from COVID-19-like disease [150].

### 2.4. DNA Vaccines

The DNA-based vaccine approach is attractive due to its easy manufacturing and convenient storage temperature and handling [151]. However, transfection of plasmid DNA is inferior to viral infection and, in contrast to mRNA-based vaccines, translocation of DNA to the cell nucleus is necessary for in vivo RNA transcription [152]. Although DNA-based vaccine candidates have been subjected to more than 500 clinical trials, only a single DNA vaccine against the H5N1 influenza virus in chicken has been approved by the USDA [151] prior to granting EUA for DNA-based COVID-19 vaccines. DNA-based vaccines are described below and summarized in Table 1. Typically, DNA-based vaccines contain the full-length SARS-CoV-2 S protein, a soluble version of it, or the RBD and a prefusion-stabilized ectodomain [153].

The synthetic INO-4800 DNA vaccine expressing the SARS-CoV-2 S elicited strong immune responses and neutralized SARS-CoV-2 in mice and guinea pigs [154]. In phase I, INO-4800 demonstrated good safety and tolerability and induced cellular and humoral immune responses [94]. Furthermore, good safety and tolerability were achieved for 1 and 2 mg doses of INO-4800 in phase II [95]. As the higher dose generated superior immunogenicity, it was selected for phase III.

In another approach, the ZyCoV-D vaccine expressing the SARS-CoV-2 S RBD induced neutralizing antibodies and Th1-biased immune responses in mice, guinea pigs, and rabbits [155]. Good safety without any serious adverse events and robust immune responses were reported in phase I/II [96]. A 66.6% vaccine efficacy was obtained in phase III [97]. EUA was granted for ZyCoV-D in India in 2021 [98].

Engineering of a synthetic soluble SARS-CoV-2 S in a plasmid generated the GX-19 vaccine [156]. Immunization of mice elicited S-specific antibody responses and protected macaques against challenges with SARS-CoV-2 [156]. Both the GX-19 vaccine and the RBD foldon, N, and S protein-containing GX-19N vaccine demonstrated good safety and tolerability profiles in phase I [99]. The GX-19 vaccine showed a superior neutralizing antibody response in comparison to the GX-19N vaccine [99].

In a different approach, saRNA DNA vectors based on the Semliki Forest virus (SFV) have been utilized for the expression of the full-length SARS-CoV-2 S protein (DREP-S) and an ectodomain of the prefusion-stabilized S protein (DREP-S-ecto) [157]. Immunization of mice with DREP-S and DREP-S-ecto induced binding and neutralizing antibodies with a superior potency detected for the former vaccine candidate. No information on clinical trials is available. 

### 2.5. RNA Vaccines

The most innovative approach to COVID-19 vaccines comprises mRNA-based approaches, where the synthetic mRNA introduced into the cytoplasm of host cells produces large quantities of antigen [158]. However, the single-stranded structure of mRNA makes it sensitive to degradation by cellular RNases, which has been addressed by the introduction of chemical modifications and anti-reverse cap analogs (ARCAs) [158]. The efficacy of mRNA delivery and protection against degradation have also been addressed by various nanoparticle (NP) formulations. Vaccine development based on mRNA has been carried out for more than 20 years, but, with the COVID-19 pandemic, the real breakthrough of the technology took place [158]. Selected mRNA-based COVID-19 vaccines are described below and summarized in Table 1.

Encapsulation of the prefusion-stabilized full-length SARS-CoV-2 S RNA into NPs resulted in the BNT162b2 vaccine [159]. Immunization of mice with BNT162b2 elicited dose-dependent antibody responses [159]. Moreover, protection against SARS-CoV-2 challenges were obtained in immunized macaques [159]. BNT162b2 showed good safety and generated robust immune responses in phase I/II [100]. In phase III, a 95% vaccine efficacy was achieved [101]. The BNT162b2 vaccine was granted EUA in the EU and Switzerland in December 2020 [102] and regulatory approval by the US FDA in August 2021 [103]. 

The mRNA-1273 vaccine is also based on the prefusion-stabilized full-length SARS-CoV-2 S RNA encapsulated in lipid nanoparticles (LNPs) [160]. The mRNA-1273 vaccine provided protection against SARS-CoV-2 in immunized mice [160] and primates [161]. In phase I, mRNA-1273 induced S-specific immune responses in all vaccinated volunteers [104]. In phase III, a 94.1% vaccine efficacy was obtained [105]. The mRNA-1273 vaccine was granted EUA in the US in December 2020 [106]. Similarly, the CVnCOV vaccine comprising LNP-encapsulated full-length SARS-CoV-2 S mRNA elicited strong humoral responses in hamsters and protected against SARS-CoV-2 challenges [162]. The immune responses were comparable to those seen in convalescent COVID-19 patients in phase I [107]. However, the vaccine efficacy was only 48.2% in phase III [108].

In contrast to the mRNA vaccines described above, the ARCoV vaccine is based on the SARS-CoV-2 S RBD encapsulated in LNPs [163]. Immunization of mice and primates induced neutralizing antibodies and Th1-biased responses [163]. Immunized mice were protected after two doses of the ARCoV vaccine [163]. An advantageous feature of the ARCoV vaccine is its thermostability [164]. The ARCoV vaccine could be stored for at least 1 week at room temperature and for 6 months at 2–8 °C without any degradation in immunogenicity [164]. In phase I, immunization with ARCoV induced robust humoral and cellular responses [109].

The application of saRNA has due to the RNA amplification step-generated superior quantities of mRNA and/or the possibility of using much lower doses of RNA for immunization [165]. Venezuelan equine encephalitis virus (VEE) replicons carrying full-length SARS-CoV-2 S RNA elicited robust S-specific antibody responses in mice and neutralized pseudovirus and wild-type SARS-CoV-2 [165]. In phase I, the LNP-nCoVsaRNA vaccine was safe, well tolerated, and elicited specific immune responses but failed to reach 100% seroconversion rates [110]. However, in phase II, superior seroconversion rates were achieved by either a prolonged dosing interval or the administration of a 1.0 μg prime dose followed by a 10 μg booster dose [111]. Another similar nanostructural lipid carrier (NLC)-encapsulated VEE-based saRNA vaccine containing the SARS-CoV-2 S RNA induced robust Th1-biased T-cell responses in immunized mice [166]. The lyophilized thermostable saRNA/NLC vaccine could be stored at refrigerated temperatures for at least 10 months. Additionally, the LUNAR-COV19 vaccine based on VEE saRNA carrying the full-length SARS-CoV-2 S RNA induced robust antibody responses and high neutralizing antibody responses and provided full protection in immunized humanized ACE2 transgenic mice [167]. 

## 3. SARS-CoV-2 Variants and Vaccine Efficacy

Although the different COVID-19 vaccines have played a significant role in reducing the COVID-19 pandemic to an endemic status in April 2023 [3,168], the emergence of SARS-CoV-2 variants has complicated the situation which can substantially reduce the efficacy of existing vaccines [169]. Emerging variants have been classified as VoC, variants of interest (VoI), and variants under monitoring (VuM) [170].

To address the reduction in vaccine efficacy due to the waning of vaccine potency, booster vaccinations with either homologous or heterologous vaccines have been conducted [171]. Additionally, intensive re-engineering of existing vaccines to especially address the mutated SARS-CoV-2 S protein is in a continuous process [172].

However, the first step has involved the testing of existing vaccines for their protection against novel variants. Obviously, it has been of major interest to verify vaccine efficacy against the most prominent variants such as alpha, beta, delta, gamma, omicron, and recent subvariants of omicron [173]. However, it is impossible in this review to describe the efficacy of all existing vaccines against each VoC. Therefore, some examples are presented below and summarized in Table 2.
viruses-16-00203-t002_Table 2Table 2Examples of COVID-19 vaccine efficacy against SARS-CoV-2 variants.Type of VaccineVariant(s)Findings**Whole-virus**

VLA2001(Valneva)delta, omicronImproved vaccine efficacy after both homologous and ChAdOx1 nCoV-19 boosters [174]**Protein and peptide**

preS dTMbeta, omicron (BA.1, BA.2, BA.4/5)Cross-reacting nAbs against VoC [38]SCB-2019alpha, beta, delta, gamma, muImmunogenicity, cross-reactivity against VoC [175]COVAX-19deltaSignificantly reduced COVID-19 rates and severity of disease [117] Nanocovaxdelta51.5% VE against delta [118]NVX-CoV2373D614G, omicron (BQ.1.1, XBB.1)High PsVNA for D614G, low for BQ.1.1 and XBB.1 after heterologous booster with NVX-CoV2373 [176]SCTV01Calpha, beta, delta, omicronPromising immunogenicity against alpha and beta, cross-reacting nAbs against delta and omicron [177]Booster elicited superior nAb titers against delta and omicron [177]UB-612delta, omicronLong-lasting nAbs, T-cell immunity against delta and omicron [178]61-fold enhanced nAbs against omicron BA.1 and 49-fold against omicron BA.2 [178].**Viral vectors**

ChAdOx1 nCoV-19Betaalpha, deltaNo protection against mild-to-moderate COVID-19 [179]Lower transmission reduction in delta than alpha, better efficacy of BNT162b2 than ChAdOx1 nCoV-19 [180]Ad26.COV2.Salpha, beta, delta, gamma, omicronbeta, gammaNeutralizing activity against VoC [181]PsVNA 5-fold (beta) and 3.3-fold (gamma) reduced [182]rAd26-S/raAd5-S(Sputnik V)Deltaalpha, beta, gamma, deltaStable VE for at least 6 months [183]Moderate (alpha) and 6.1- (beta) [184], 2.8- (gamma) [185], 2.5-fold (delta) [185] reduction in nAb activityNDV-HXP-Sbeta, delta, omicronProtection against beta, delta, cross-nAbs against omicron [186]Flu-RBD VLPsdeltaInduced nAbs [187]VSV-ΔG-S(BriLife^®^)alpha, beta, gamma, delta, omicronComparable levels of nAbs for alpha, gamma, delta, 3-fold reduction for beta and omicron [150]**Nucleic acid—DNA**

INO-4800alpha, beta, gamma2.1-fold (alpha), 6.9-fold (beta) reduction in nAbs, same as for the original strain (gamma) [188]INO-4802alpha, beta, gammanAb responses [189], prime-boost strategy [190]**Nucleic acid—RNA**

BNT162b(Pfizer/BioNTech)alpha, beta, gamma, deltadeltaomicronPrevention of symptomatic and severe COVID-19 [191]67% (Spain) [192] and 90% (the UK) [193] VE25% (US) [194], 30% (Israel) [195], 51% (Qatar) [196] VEmRNA-1273(Moderna)alpha, beta, gamma, delta1.2-fold (alpha), 2.1–8.4-fold (beta, gamma, delta) nAb titer reduction [197]mRNA-1273.214Omicronomicron BA.1, BA.4, BA.5alpha, beta, gamma, deltaReduced nAbs in children compared to D614G [198]Superior nAb response compared to mRNA-1273 [199]Superior binding compared to mRNA-1273 [199]mRNA XBB.1.5omicron XBB.1.5, EG.1.527–27.6-fold increase in nAb levels [200]CVnCOV(Curavec)deltaSimilar nAb levels as for wild-type SARS-CoV-2 [201]LNP-nCoVsaRNAalphaProtection against alpha variant in hamsters [202]ZIP1642beta, deltaSubstantial nAb responses [203]saRNA/NLC SARS-CoV-2alpha, beta, deltaRobust nAb and Th1-biased T-cell responses [166]Flu, inactivated influenza A virus; nAbs, neutralizing antibodies; LNP, lipid nanoparticles; NDV, Newcastle disease virus; NLC, nanolipid particles; PsVNA, pseudovirus-neutralizing antibody; RBD, receptor binding domain; saRNA, self-amplifying RNA; VE, vaccine efficacy; VLP, virus-like particle; VoC, variant of concern.

For example, the efficacy of the whole-virus vaccine VLA2001 against the delta and omicron variants was improved after both homologous and ChAdOx1 nCoV-19 heterologous booster immunizations [174]. In another study, it was demonstrated that individuals originally vaccinated with mRNA or Ad-based vaccines, who received a booster immunization with the preS dTM-AS03 protein subunit vaccine carrying the beta variant S protein, showed robust neutralizing antibody responses against several SARS-CoV-2 VoC [38]. The booster formulation induced cross-reacting neutralizing antibodies against the beta (B.1.351) and omicron (BA.1, BA.2, and BA.4/5) variants [38]. It has also been shown that the SCB-2019 protein subunit vaccine induced immunogenicity and cross-reactivity against the alpha, beta, gamma, delta, and mu variants in phase II/III [175]. In addition, vaccination with the COVAX-19 vaccine based on the S protein ECD resulted in significantly reduced COVID-19 rates and severity of disease caused by the delta variant [116]. Moreover, the Nanocovax vaccine showed good safety and a 51.5% vaccine efficacy against the delta variant [117]. The NVX-CoV2373 protein subunit vaccine demonstrated the highest pseudovirus-neutralizing antibody (PsVNA) responses for the SARS-CoV-2 D614G variant and the lowest PsVNA responses for the omicron BQ.1.1 and XBB.1 variants after heterologous booster vaccination of individuals previously vaccinated with Ad26.COV2.S, mRNA-1273, or BNT162b2 vaccines [176]. In the case of the SCTV01C based on the SARS-CoV-2 S ECD, promising immunogenicity against the alpha and beta variants and cross-neutralizing antibodies against the delta and omicron variants were obtained in phase I [50]. Furthermore, booster vaccinations with SCTV01C of individuals previously vaccinated with the whole-virus vaccine BBIBP-CorV resulted in superior neutralizing antibody titers against the delta and omicron variants in phase III [177]. In the case of the S protein epitope-based UB-612 vaccine, long-lasting neutralizing antibody responses were induced and a broad T-cell immunity against the delta and omicron variants was detected in phase I/II [178]. Moreover, booster vaccinations with UB-612 enhanced the neutralizing antibody levels by 131-, 61-, and 49-fold against the ancestral SARS-CoV-2 strain and the omicron BA.1 and BA.2 variants, respectively [178].

Related to viral vector-based vaccines, ChAdOx1 has been evaluated in a two-dose regimen against the beta variant in pseudovirus and live-virus neutralization assays [179]. The study showed that the vaccination was unable to provide protection against mild-to-moderate COVID-19 caused by the beta variant. In another study, the association between transmission of the alpha and delta variants and vaccination with either the ChAdOx1 nCoV-19 vaccine or the BNT162b2 mRNA vaccine was evaluated [180]. The effect of vaccination on reduced transmission was inferior for the delta variant compared to the alpha variant. Moreover, the mRNA-based vaccine provided a greater decrease in transmission than the Ad-based ChAdOx1 nCoV-19 vaccine. The ChAdOx1 nCoV-19 vaccine showed, in another study, neutralizing activity against the alpha, beta, delta, gamma, and omicron variants [203]. In the case of the single-dose Ad26.COV2.S vaccine, neutralizing antibody responses were reduced against the beta and gamma variants compared to the original SARS-CoV-2 strain [181]. In another study, it was demonstrated that the vaccine efficacy against the delta variant was stable for at least 6 months [182]. The Ad-based Sputnik V vaccine showed a moderate decrease in the neutralization activity of a VSV-based pseudovirus for the alpha variant, whereas the reduced activity for the beta variant was 6.1-fold [183]. In the case of the gamma [184] and delta [204] variants, 2.8- and 2.5-fold reduced neutralization activity was detected, respectively.

In the case of NDV-based vaccines, the next-generation trivalent NDV-HXP-S vaccines expressing the prefusion-stabilized S protein of the original SARS-CoV-2 strain, the gamma and delta variants significantly enhanced the protection against disease and provided cross-neutralizing antibodies against distant variants such as omicron [185]. In another approach, the inactivated influenza A virus (Flu) was conjugated with the recombinant SARS-CoV-2 S RBD, which generated SARS-CoV-2 virus-like particles (VLPs) [186]. The Flu-RBD VLPs induced strong neutralization activity against both wild-type SARS-CoV-2 and the delta variant. 

In an approach to address the potential vaccine escape of VoC, the replication-competent VSV-ΔG-S (BriLife^®^) has been utilized for the expression of S mutations corresponding to those present in VoC [150]. It was demonstrated that the VSV-ΔG-S vaccine elicited comparable neutralizing antibody levels against the alpha, gamma, and delta variants and a 3-fold reduction against beta and omicron compared to the original SARS-CoV-2 strain. 

In the case of DNA vaccines, the INO-4800 vaccine showed comparable levels of neutralizing antibodies for the gamma variant and the original SARS-CoV-2 strain and 2.1-fold and 6.9-fold reduced levels of neutralizing antibodies for the alpha and beta strains, respectively [187]. To better address the vaccine efficacy against emerging variants, the pan-SARS-CoV-2 DNA vaccine INO-4802 was designed [188]. INO-4802 elicited potent neutralizing antibodies and T-cell responses against the original SARS-CoV-2 strain and the alpha, beta, and gamma variants. Moreover, a prime-boost regimen with INO-4802 enhanced and broadened the immune response in rhesus macaques [189]. 

Related to mRNA-based vaccines, the BNT162b2 vaccine showed efficacy in preventing symptomatic COVID-19 and severe disease against the alpha, beta, gamma, and delta variants [190]. In other studies, 67% and 90% efficacy against the delta variant was reported for vaccinations with BNT162b2 in Spain [191] and the UK [192], respectively. In the case of the omicron variant, 25% vaccine efficacy was discovered in the US [193], 30% in Israel [194], and 51% in Qatar [195] against the omicron BA.1 and BA.2 variants. In the case of the mRNA-1273 vaccine, a 1.2-fold statistically non-significant reduction of neutralizing antibody titers was obtained against the alpha variant in comparison to the D614G strain [196]. In contrast, a 2.1- to 8.4-fold reduction of titers was seen for the beta, gamma, and delta variants [196]. In another study, the mRNA-1273 vaccine induced neutralizing antibodies against omicron in both adolescents and children although at lower levels compared to the D614G strain [197]. A comparative phase II/III study of booster vaccinations was conducted with the original mRNA-1273 and the bivalent novel mRNA-1273.214 vaccine comprising the original mRNA-1273 and the omicron BA.1 S RNA [198]. The bivalent vaccine elicited neutralizing antibody titers against the omicron BA.1 of 2372.4 compared to 1473.5 for the original mRNA-1273 vaccine. Moreover, mRNA-1273.214 induced neutralizing antibody titers of 727.4 and 492.1 against the omicron BA.4 and BA.5 variants, respectively. The bivalent vaccine also induced higher binding activity against the alpha, beta, gamma, and delta variants than the mRNA-1273 vaccine [198]. The mRNA XBB.1.5 vaccine booster elicited 27-fold and 27.6-fold enhanced neutralizing antibody levels against the omicron XBB.1.5 and EG5.1 variants [199]. In the case of the CVnCoV mRNA vaccine, a third dose elicited neutralizing antibody responses against both the SARS-CoV-2 wild-type and the delta variant [200]. The antibody levels were higher than those obtained after two vaccinations.

In the context of saRNA-based COVID-19 vaccines, the LNP-nCoVsaRNA vaccine elicited strong immune responses and protected immunized hamsters against the D614G strain and the alpha variant [201]. Another saRNA-based vaccine, the ZIP1642 saRNA encoding both the SARS-CoV-2 S RBD and N RNA, elicited high neutralizing antibody titers against the beta and delta variants [202]. Moreover, the thermostable saRNA/NLC SARS-CoV-2 vaccine elicited robust neutralizing antibody responses and Th1-biased T-cell responses against the alpha, beta, and delta variants [166]. 

Comparative studies on current COVID-19 vaccines have been carried out. For example, T-cell responses against the alpha, beta, gamma, kappa, delta, lambda, and omicron variants were obtained after vaccinations with Ad26.COV2.S, NVX-CoV2373, mRNA-1273, and BNT162b2 [175]. The study demonstrated that the cross-reactivity against early variants (alpha and beta) was preserved for the different vaccines. In the case of omicron, memory T-cell responses were also preserved. However, the memory B cells and neutralizing antibodies were reduced to 42%. In a systematic review and meta-analysis, 11 COVID-19 vaccines were evaluated for vaccine efficacy against the alpha, beta, gamma, delta, and omicron variants [205]. Full vaccinations against all variants were effective, ranging between 88.0 and 55.9%. The booster vaccinations were superior for the delta and omicron variants. The mRNA-based COVID-19 vaccines generated better vaccine efficacy against the alpha, beta, gamma, and delta variants compared to the Ad- and protein-based vaccines. Not enough evidence was available for the evaluation of the omicron variant at the time of publication [205]. 

Finally, a summary of patients with pre-existing medical conditions and COVID-19 vaccines is presented below. For example, as an increased risk of severe COVID-19 and a higher mortality rate have been seen in cancer patients, a study was carried out in patients with solid or hematological cancers [206]. A comparison was performed of the S-RBD IgG levels and neutralizing antibody levels in individuals who had received two doses of the BNT162b2 or mRNA-1273 RNA vaccines, the BBIBP-CorV whole-virus vaccine, or the ChAdOx1 nCoV-19 vaccine within 6 months. The study revealed that, even though immune responses were detected in all cancer patients, the anti-SARS-CoV-2 S-RBD IgG and neutralizing antibody levels were significantly higher after vaccinations with the mRNA vaccines than with non-mRNA vaccines. In another study, patients with solid tumors subjected to a third dose of the BNT162b2 mRNA vaccine were evaluated for humoral and cellular immune responses [207]. The booster vaccination induced significant humoral and cellular responses in cancer patients although the neutralizing antibody levels were lower against the omicron variant [207]. Moreover, immunosuppressed dermatological patients who received the inactivated CoronaVac vaccine responded poorly despite having low-level immunosuppression [208]. Furthermore, it was revealed that patients receiving azathioprine, cyclosporin mycophenolate mofetil, or prednisolone showed lower levels of anti-SARS-CoV-2 IgG and neutralizing antibodies than those who were treated with methotrexate. In another approach, humoral responses and the severity of COVID-19 after a fourth booster vaccination with either the BNT16b2 or ChAdOx1 nCoV-19 vaccines were compared in hemodialysis patients [209]. It was concluded that the booster vaccination generated significantly elevated antibody levels in hemodialysis patients, which was beneficial in protection against severe COVID-19.

## 4. Conclusions and Future Aspects

Various COVID-19 vaccines based on whole viruses, protein subunits and peptides, viral vectors, and nucleic acids have demonstrated efficacy in protecting against SARS-CoV-2 infections and reducing the severity of COVID-19. Vaccine efficacy in the 90–95% range has been achieved in clinical evaluations for different vaccine types [69,74,101,105]. Moreover, EUA has been granted for COVID-19 vaccines in a large number of countries around the world [14,18,33,34,51,55,62,67,70,75,78,81,98,102,106], which led to unprecedented global mass vaccinations. As of the end of November 2023, a total of 13.6 billion vaccine doses had been administered globally, and more than 5.2 billion individuals had received a primary series of vaccines [210]. The vaccination strategy significantly contributed to the downgrading of the pandemic to an endemic status [2]. However, mass vaccinations presented major issues related to the logistics of distribution, storage, and administration on a global scale, especially in developing countries. Not surprisingly, the administration of billions of doses revealed adverse events never accounted for before including cases of vaccine-induced thrombotic thrombosis (VITT) [211]. Despite numerous efforts of unfounded statements, disinformation, and conspiracy theories indicating major danger and serious consequences of vaccinations, scientific evidence has clearly confirmed that the advantages of COVID-19 vaccines strongly outweigh the risks of SARS-CoV-2 and severe disease.

Another issue of COVID-19 vaccines relates to the waning of vaccine potency with time, which has been addressed by booster vaccinations with both homologous and heterologous vaccines [169]. Moreover, the emergence of SARS-CoV-2 variants, classified as VoC, VoI, and VuM, has hampered the potency of existing vaccines [4]. This has triggered the engineering of bivalent [212] and novel variant-specific vaccines [172]. Several studies, as described above and summarized in Table 2, have confirmed that the vaccine efficacy is reduced for most of the variants, especially for the most recent omicron variants. However, booster vaccinations can enhance immune responses and thereby provide better protection against COVID-19 and also reduce the severity of the disease. Evidently, both bivalent and variant-specific vaccines have proven superior to the original vaccines.

In general, vaccine development seems to be a race between emerging variants and the re-engineering of novel vaccines specifically targeting those modifications in the SARS-CoV-2 S protein enabling immune escape. However, alternative approaches of targeting other SARS-CoV-2 proteins than the S protein such as the engineering of the pan-immune responses for whole-virus vaccines [5] and the INO-4802 pan-SARS-CoV-2 vaccine [188] have been tested. Another improvement relates to the development of thermostable vaccines such as the saRNA/NLC [167] and saRNA/NLC SARS-CoV-2 vaccines [166], which facilitate transport and storage but should also improve vaccine stability upon administration.

An important issue related to COVID-19 vaccine development deals with the re-engineering of novel vaccine versions to address the reduced efficacy of existing vaccines against emerging SARS-CoV-2 variants. In this context, older COVID-19 vaccines have been withdrawn and approval has been granted for re-engineered versions. The FDA has simplified COVID-19 vaccine schedules facilitating the withdrawal of older COVID-19 vaccines and supporting the approval of updated bivalent and VoC-targeting vaccines [213]. For example, the FDA withdrew the EUA for the original mRNA-1273 and the bivalent vaccines and replaced them with newer omicron-targeted vaccines in June 2023 [214]. Moreover, the bivalent Moderna and Pfizer-BioNTech mRNA vaccines were withdrawn in September 2023 by the FDA to support updated mRNA vaccines, which can provide better protection against current SARS-CoV-2 variants [215]. In May 2023, Janssen requested the FDA to withdraw its Ad26.COV2.S vaccine, which was not associated with concerns about its quality or safety [216]. In October 2023, the FDA amended the EUA for Novavax and an updated version including the spike protein from the omicron XBB.1.5 variant has been engineered [217].

In summary, the experience from the COVID-19 pandemic and the awareness of the necessity to sustain vaccine efficacy against emerging SARS-CoV-2 variants put the world in a better place to meet the challenges of facing potential new outbreaks to prevent new pandemics [208]. 

## Data Availability

No research data were generated in this review.

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
