# Peer review of "COVID-19 Vaccines: Where Did We Stand at the End of 2023?"

_viruses, 2024, doi:10.3390/v16020203_

Round 1

Reviewer 1 Report

Comments and Suggestions for Authors

The topic is very interesting, but it should examine, even briefly, the various types of vaccines in "special" populations (e.g. PMID: 37529941, PMID: 36857323,PMID: 34957149, PMID: 36029652,PMID: 37896987)Please expand this review by adding a paragraph on this topic, using the listed bibliographic entries and adding relevant ones

Comments on the Quality of English Language

None

Author Response

The topic is very interesting, but it should examine, even briefly, the various types of vaccines in "special" populations (e.g. PMID: 37529941, PMID: 36857323, PMID: 34957149, PMID: 36029652, PMID: 37896987). Please expand this review by adding a paragraph on this topic, using the listed bibliographic entries and adding relevant ones.

Response: A paragraph has been added as suggested. However, the first reference (PMID: 37529941) has not been excluded as the study is conducted on 32 healthy volunteers.

Reviewer 2 Report

Comments and Suggestions for Authors

The author thoroughly covers all types of COVID-19 vaccines and from a world perspective which is very much appreciated and needed.  They provide valuable global coverage of the vaccines designed and tested. Overall, there are some major issues with the references as it appears in several parts that the reference have shifted and are not longer cited correctly.   

The tables although informative could have been even better if placed in columns with % efficacy in a separate column to quickly identify and compare.

Abstract needs modified. 

Line 11 talks about safety but in the text it is barely a passing thought.  Would eliminate “although…. Vaccination” and then add in line 15 after variants “, high safety in a global mass vaccination campaign with rare cases of adverse events,”.

Line 15  states “other future viral infection” either insert SARS-CoV-2 or eliminate as none are provided.

Introduction

Lines 22-25.  Looks like a paste error. I believe you were trying to state: “Rapid development of vaccines and mass vaccination with efficient COVID-19 vaccines contributed to the pandemic being downgraded to an endemic status in spring of 2023.”

Line 28-32 needs to be changed.  Maybe ….authorities “combined” delete “managed” to demonstrate the safety….

Line 36-37 needs to be changed.  At the end add “ are needed” or maybe “are emerging”.

Line 51 needs “,” between “vaccines” and “the”

Line 59 – reference 9 does not seem appropriate as I would expect an actual covi-vac study to be referenced and not an innovator review.

Line 123 reference 42 is wrong – describes psueudotumor cerebri in neurology

Line 154 reference 64 the text is the Granting of EUA in China and Uzbekistan yet reference describes booster vaccine does not seem appropriate.

Line 164. Reference 72 is for India and Botswana EUA yet reference only is for India…need a reference for Botswana.

Line 177 need to add “in mice and NHP” as it reads as human reference.

Line 224, reference 104 is not correct. It should be “Food and Drug Administration. Fact sheet for healthcare providers administering vaccine (vaccination providers) Emergency Use Authorization (EUA) of the Janssen COVID-19 vaccine to prevent coronavirus disease (COVID-19). Silver Spring, MD: Food and Drug Administration; 2019. https://www.fda.gov/media/146304/download

Line 325 Reference 144 seems an odd choice to support the most innovative approach for mRNA. Confirm that it is correct.

Line 428-429 only a reference for ChAdOx1 is present but should also have one for BNT162b2. Please insert.

Line 437 Reference 183 belongs with the sentence following its insertion (Ad-based Sputnik V) not with the Ad26.COV2.S.  Fix.

Line 463 NHP study is reference from a review 191 not from a direct article.  Double check to see if there is any article on BioRX.

Line 469. Reference 196 is a reference for Qatar not Israel.  Provide an Israel reference please.

Line 469 Reference 197 is not a Qatar reference fix please.

Line 485 Reference 201 is used to reference Omicron variant but the reference is for Delta and and wild type.  Need a different reference.

Line 486 Reference 202 is used for delta VOC but reference contains no Delta data.  Please fix.

Line 492  Reference 204. Indicates a spike RBD and Nucleocapsid RNA, within the paper there is no mention of Nucleocapsid thus the sentence or the reference needs fixed.

Line 493 Reference 205 should be 204 as 205 does not contain saRNA/NLC SARS vaccines.

Line 503  Reference 206 WHO dashboard does not take you to the 11 vaccines only the cases, please update

Line 508  Reference 206 WHO dashboard seems off

Line 518 Reference 207 is incorrect, I believe you meant 206 for this.

Line 523 Reference 207 does not seem appropriate for VITT.

Line 543  after reference 5 remove “..”

Comments on the Quality of English Language

Overall the English is well done; however, there appears to be some editing issues in the introduction which had several sentences that were not sound.  I have made suggestions.

Author Response

The author thoroughly covers all types of COVID-19 vaccines and from a world perspective which is very much appreciated and needed.  They provide valuable global coverage of the vaccines designed and tested. Overall, there are some major issues with the references as it appears in several parts that the references have shifted and are no longer cited correctly.   

Response: The manuscript has been revisited and the references checked, replaced or updated as suggested.

The tables although informative could have been even better if placed in columns with % efficacy in a separate column to quickly identify and compare.

Response: Although I appreciate this suggestion, I am not sure how valuable it would be to introduce a separate column for vaccine efficacy.4

 The abstract needs to be modified. 

Response: The Abstract section has been revised.

Line 11 talks about safety but in the text it is barely a passing thought.  Would eliminate “although…. Vaccination” and then add in line 15 after variants “, high safety in a global mass vaccination campaign with rare cases of adverse events,”.

Response: “I respectfully disagree with the comment and suggest that the text should remain in its current form.

Line 15  states “other future viral infection” either insert SARS-CoV-2 or eliminate as none are provided.

Response: The sentence has been revised accordingly

Introduction

Lines 22-25.  Looks like a paste error. I believe you were trying to state: “Rapid development of vaccines and mass vaccination with efficient COVID-19 vaccines contributed to the pandemic being downgraded to an endemic status in spring of 2023.”

Response: The sentence has been revised accordingly.

Line 28-32 needs to be changed.  Maybe ….authorities “combined” delete “managed” to demonstrate the safety….

Response: The sentence has been revised accordingly

Line 36-37 needs to be changed.  At the end add “ are needed” or maybe “are emerging”.

Response: The sentence has been revised accordingly

Line 51 needs “,” between “vaccines” and “the”

Response: The sentence has been revised accordingly

Line 59 – reference 9 does not seem appropriate as I would expect an actual covi-vac study to be referenced and not an innovator review.

Response: Reference 9 has been replaced.

Line 123 reference 42 is wrong – describes psueudotumor cerebri in neurology

Response: Reference 9 has been replaced.

Line 154 reference 64 the text is the Granting of EUA in China and Uzbekistan yet reference describes booster vaccine does not seem appropriate.

Response: Reference 64 has been replaced and an additional reference (65) has been added for the approval in Uzbekistan.

Line 164. Reference 72 is for India and Botswana EUA yet reference only is for India…need a reference for Botswana.

Response: Reference 74 has been added for the approval in Uzbekistan.

Line 177 need to add “in mice and NHP” as it reads as human reference.

Response: I do not understand this comment as in the text it explicitly says, “achieved in mice and non-human primates”.

Line 224, reference 104 is not correct. It should be “Food and Drug Administration. Fact sheet for healthcare providers administering vaccine (vaccination providers) Emergency Use Authorization (EUA) of the Janssen COVID-19 vaccine to prevent coronavirus disease (COVID-19). Silver Spring, MD: Food and Drug Administration; 2019. https://www.fda.gov/media/146304/download”

Response: Reference 104 (now 106) has been replaced as suggested.

Line 325 Reference 144 seems an odd choice to support the most innovative approach for mRNA. Confirm that it is correct.

Response: Reference 144 (now 146) has been replaced as suggested.

Line 428-429 only a reference for ChAdOx1 is present but should also have one for BNT162b2. Please insert.

Response: The cited publication should be the reference 179 (now 181) instead of 180, which has now been corrected.

Line 437 Reference 183 belongs with the sentence following its insertion (Ad-based Sputnik V) not with the Ad26.COV2.S.  Fix.

Response: The reference has been corrected.

Line 463 NHP study is reference from a review 191 not from a direct article.  Double check to see if there is any article on BioRX.

Response: The correct reference 190 (now 192) has been cited.

Line 469. Reference 196 is a reference for Qatar not Israel.  Provide an Israel reference please.

Response: The correct reference for Israel is 195 (now 197) and indeed for Qatar 196 (now 198). This has now been corrected.

Line 469 Reference 197 is not a Qatar reference fix please.

Response: The correct reference 196 (now 198) has been introduced.

Line 485 Reference 201 is used to reference Omicron variant but the reference is for Delta and and wild type.  Need a different reference.

Response: The reference has been changed to  200 (now 201).

Line 486 Reference 202 is used for delta VOC but reference contains no Delta data.  Please fix.

Response: The reference has been changed to 200 (now 201).

Line 492  Reference 204. Indicates a spike RBD and Nucleocapsid RNA, within the paper there is no mention of Nucleocapsid thus the sentence or the reference needs fixed.

Response: The reference has been changed to 203 (now 205).

Line 493 Reference 205 should be 204 as 205 does not contain saRNA/NLC SARS vaccines.

Response: The reference has been changed to 204 (now 206).

Line 503  Reference 206 WHO dashboard does not take you to the 11 vaccines only the cases, please update

Response: The reference has been changed to 205 (now 207).

Line 508  Reference 206 WHO dashboard seems off

Response: The reference has been replaced now.

Line 518 Reference 207 is incorrect, I believe you meant 206 for this.

Response: The reference has been changed to 206 (now 208).

Line 523 Reference 207 does not seem appropriate for VITT.

Response: The reference has been replaced by a more appropriate one.

Line 543  after reference 5 remove “..”

Response: The correction has been done.

Comments on the Quality of English Language

Overall the English is well done; however, there appears to be some editing issues in the introduction which had several sentences that were not sound.  I have made suggestions.

Response: The manuscript has been revised accordingly taking into account your valuable comments when possible.

Reviewer 3 Report

Comments and Suggestions for Authors

The review provides a comprehensive overview of the successful development of COVID-19 vaccines against SARS-CoV-2, highlighting various approaches and demonstrating good safety and efficacy in preclinical and clinical studies. It acknowledges the rare cases of severe adverse events post-vaccination and addresses the impact of emerging SARS-CoV-2 variants on vaccine efficacy, emphasizing the need for redesigning and re-engineering novel COVID-19 vaccine candidates. The inclusion of insights into preparedness for future viral infections adds valuable context to the discussion. The overall writing by the author is coherent; however, sudden breaking of sentences by period can be avoided. Two smaller sentences can be joined together using conjunctions to keep the flow while reading. Since the title states 'end of 2023,' it would be beneficial to add more examples of vaccines that were approved or withdrawn by the end of 2023. Additionally, it would be more helpful to add details of updated formula vaccines along with the ones already mentioned in the text, i.e., 'October 3, 2023, FDA authorized the updated Novavax COVID-19 vaccine'. In a similar vein it would be beneficial to add details on such vaccines that were first authorized but later removed for use in countries like the USA i.e. Novavax.

Comments on the Quality of English Language

The review provides a comprehensive overview of the successful development of COVID-19 vaccines against SARS-CoV-2, highlighting various approaches and demonstrating good safety and efficacy in preclinical and clinical studies. It acknowledges the rare cases of severe adverse events post-vaccination and addresses the impact of emerging SARS-CoV-2 variants on vaccine efficacy, emphasizing the need for redesigning and re-engineering novel COVID-19 vaccine candidates. The inclusion of insights into preparedness for future viral infections adds valuable context to the discussion. The overall writing by the author is coherent; however, sudden breaking of sentences by period can be avoided. Two smaller sentences can be joined together using conjunctions to keep the flow while reading. Since the title states 'end of 2023,' it would be beneficial to add more examples of vaccines that were approved or withdrawn by the end of 2023. Additionally, it would be more helpful to add details of updated formula vaccines along with the ones already mentioned in the text, i.e., 'October 3, 2023, FDA authorized the updated Novavax COVID-19 vaccine'. In a similar vein it would be beneficial to add details on such vaccines that were first authorized but later removed for use in countries like the USA i.e. Novavax.

Author Response

The review provides a comprehensive overview of the successful development of COVID-19 vaccines against SARS-CoV-2, highlighting various approaches and demonstrating good safety and efficacy in preclinical and clinical studies. It acknowledges the rare cases of severe adverse events post-vaccination and addresses the impact of emerging SARS-CoV-2 variants on vaccine efficacy, emphasizing the need for redesigning and re-engineering novel COVID-19 vaccine candidates. The inclusion of insights into preparedness for future viral infections adds valuable context to the discussion. The overall writing by the author is coherent; however, sudden breaking of sentences by period can be avoided. Two smaller sentences can be joined together using conjunctions to keep the flow while reading.

Response: The manuscript has been thoroughly edited and revised.

Since the title states 'end of 2023,' it would be beneficial to add more examples of vaccines that were approved or withdrawn by the end of 2023. Additionally, it would be more helpful to add details of updated formula vaccines along with the ones already mentioned in the text, i.e., 'October 3, 2023, FDA authorized the updated Novavax COVID-19 vaccine'. In a similar vein it would be beneficial to add details on such vaccines that were first authorized but later removed for use in countries like the USA i.e. Novavax.

Response: A paragraph has been added on examples of withdrawal of old vaccines and the approval of their replacements.

Reviewer 4 Report

Comments and Suggestions for Authors

The masnucript provides comprehensive review of COVID-19 vaccines both authorised or under devolpment. 

- The text might benefit from providing the time context to the results of clinical trials because the evolution of SARS-CoV-2 variants and subvariants significantly affects the efficiency of vaccination or alternatively may provide the insight about predominat variant at the time of clinical trial realisation. 

Author Response

The manuscript provides a comprehensive review of COVID-19 vaccines both authorised or under development. 

- The text might benefit from providing the time context to the results of clinical trials because the evolution of SARS-CoV-2 variants and subvariants significantly affects the efficiency of vaccination or alternatively may provide the insight about predominant variant at the time of clinical trial realisation. 

Response: Including a chronological listing of the clinical trials would require some major effort and would add a substantial number of numerical values to the manuscript. The timelines for most clinical trials are fairly obvious from the granting of EUA listed in Table 1. Additionally, Table 2 lists the essential VoC at the time of various clinical trials.